# The *pcnB* gene sustains *Shigella flexneri* virulence

Thibault Frisch[1], Petra Geiser[1], Margarita Komi[2], Philip A. Karlsson[1], Anjeela Bhetwal[1,2], Laura Jenniches[3], Lars Barquist[3,4,5], Erik Holmqvist[2], André Mateus[6,7], Mikael E. Sellin[1,8], Maria Letizia Di Martino[1]*

1 Department of Medical Biochemistry and Microbiology, Uppsala University, Uppsala, Sweden, 2 Department of Cell and Molecular Biology, Uppsala University, Uppsala, Sweden, 3 Helmholtz Institute for RNA-based Infection Research (HIRI), Helmholtz Centre for Infection Research (HZI), Würzburg, Germany, 4 Department of Biology, University of Toronto, Mississauga, Ontario, Canada, 5 Department of Cell and Systems Biology, University of Toronto, Toronto, Ontario, Canada, 6 Department of Chemistry, Umeå University, Umeå, Sweden, 7 The Laboratory for Molecular Infection Medicine Sweden (MIMS), Umeå University, Umeå, Sweden, 8 Science for Life Laboratory, Uppsala, Sweden

* ml.dimartino@imbim.uu.se

## Abstract

The enteropathogen *Shigella flexneri* employs a Type Three Secretion System (T3SS) to colonize intestinal epithelial cells. Genes encoding the T3SS are located on a large IncFII virulence plasmid, pINV. T3SS expression comes at the expense of slowed *Shigella* growth and is therefore strictly controlled by both transcriptional and post-transcriptional mechanisms. Following up on a recent genome-wide screen, we here show that the chromosomal gene *pcnB,* encoding the poly-A polymerase I (PAP-I), slows *Shigella* growth at 37°C, while it at the same time promotes early colonization of a human epithelial enteroid model. Proteomic profiling revealed that *pcnB* drives global increase of the *Shigella* T3SS virulence program. Accordingly, *pcnB* upholds pINV replication to a level favourable for *Shigella* virulence. This is achieved through increased degradation of the antisense RNA CopA, involved in plasmid replication control. The *pcnB* effect on pINV replication was found to also ensure longer-term intraepithelial expansion of *Shigella* following human intestinal epithelium invasion. Our findings exemplify how an adequate pINV level, sustained by *pcnB*, underpins the successful execution of *Shigella*´s infection cycle.

## Author summary

Bacterial infections represent a major global threat. Understanding the genetic determinants promoting infections is crucial to overcome this threat. *Shigella* is an intracellular bacterial pathogen that invades and disseminates in the intestinal epithelium, causing bacillary dysentery in humans. *Shigella*´s ability to cause disease relies on the delivery of effector proteins into the host cells through an

**Data availability statement:** The mass spectrometry proteomics data have been deposited at the ProteomeXchange Consortium via the PRIDE partner repository and will be available with the dataset identifier PXD046629.

**Funding:** This work was supported by grants from the Carl Trygger Foundation (CTS 22:1915 to M.L.D.M.), the Swedish Research Council (2018-02223, 2022-01590 to M.E.S.), and the SciLifeLab Fellows program (to M.E.S.). The proteomics analysis performed in the Mateus laboratory was enabled by a grant from Kempestiftelserna (JCK3126 to A.M.). The funders had no role in study design, data collection and analysis, decision to publish, or preparation of the manuscript.

**Competing interests:** The authors have declared that no competing interests exist.

injection machinery, with most of the genes involved in this process located on a large virulence plasmid. Here we show that the chromosomal gene *pcnB* sustains a suitable virulence plasmid level. This is crucial for *Shigella* to maximize virulence protein expression and thereby efficiently invade, replicate and spread within the intestinal epithelium.

## Introduction

Virulence represents the ability of a bacterium to infect and inflict damage to a host [1]. Numerous intracellular bacterial pathogens inject virulence effector proteins through dedicated secretion systems into mammalian host cells in order to colonize and cause disease [2]. *Shigella flexneri* (hereafter *Shigella*), a primate-restricted intracellular pathogen causing bacillary dysentery, delivers multiple effectors into non-phagocytic host cells through a Type Three Secretion System (T3SS), resulting in bacterial uptake, spread to adjacent cells, immune evasion and ultimately inflammatory destruction of infected gut mucosal regions [reviewed in 3,4].

Genes encoding *Shigella* T3SS, effectors, and chaperones are located on a large virulence plasmid (pINV) of ~232 kb organized into two divergently transcribed operons (*ipa* and *mxi-spa*) within a conserved pathogenicity island (PAI) of 32 kb [5–8]. This entry region also contains *virB* and *mxiE* genes, encoding transcriptional activators required for the expression of the majority of pINV virulence genes [9,10]. Other genes encoding proteins crucial for the invasive process are scattered around the pINV. This includes *icsA*, involved in actin-based intraepithelial spread [11], several effectors involved in immune evasion (e.g., *osp* and *ipaH* effectors; [3]) and *virF,* encoding the master virulence regulator [12–17]. pINV harbours a RepFIIA replicon, containing the origin of replication (ori) and the gene encoding the replication initiation protein RepA. The *tap* gene, located upstream of *repA*, encodes a short leader peptide and its translation is coupled to *repA* translation [7]. Replication control of IncFII plasmids has been extensively studied using the R1 plasmid from *Salmonella enterica* as a model [18]. IncFII plasmid replication is mainly governed by the antisense sRNA CopA, which inhibits RepA translation by preventing translation of Tap [18–20]. CopA expression is constitutive and results in a highly unstable transcript, characterized by a very short half-life. Therefore, CopA concentration reflects the plasmid copy number in the bacterial population [21,22]. A second replication repressor, the protein CopB (also known as RepA2 in *Shigella*) acts as a quadrimer and inhibits *repA* transcription by binding to its promoter region [23–25]. *copB* transcription is also constitutive, so that high CopB levels derived from a high plasmid copy number inhibit plasmid replication [26]. Virulence plasmids across several bacterial species belong to the IncFII incompatibility group [7]. *Shigella* pINV, moreover, harbours two redundant partitioning systems, ParA/B and StbA/B [27], as well as three different Toxin-Antitoxin systems (MvpAT also known as VapBC, GmvAT, CcdAB) to ensure pINV maintenance and stability across different conditions [7,27].

Expression of the T3SS and related effectors comes at a significant growth cost and is therefore tightly regulated in response to several environmental stimuli, such as temperature, pH and osmolality. In particular, shift to 37°C, the host temperature, is crucial to promote virulence gene expression. The key event following the upshift of *Shigella* to 37°C is the translation of VirF, which activates expression of the *icsA* and *virB* genes [12,15–17,28–30]. IcsA is involved in *Shigella* spread between adjacent cells through actin polymerization of the host [11,31], while VirB in turn activates the T3SS structural component genes (Mxi and Spa proteins), its early effectors (Ipa and Ipg proteins) and their chaperones, as well as the most downstream regulator in the cascade, MxiE [32]. Once *Shigella* senses the host environment through T3SS contact with the host cells, MxiE activates the expression of late effector proteins, mainly involved in immune evasion [32].

We have recently mapped the genome-wide cost of virulence gene expression in *Shigella*, employing a functional genomics approach. In parallel, we also established a genome-wide map of *Shigella* genes required for early colonization of the human intestinal epithelium, using physiologically relevant human enteroids, mini replicas of the gut, as a model [33]. Here, following up on this previous work, we found that the *pcnB* gene (plasmid copy number B), encoding the polyadenylase poly-A polymerase I (PAP-I) [34], is costly for *Shigella* growth at 37°C (virulence on), while it boosts the bacterium´s ability to colonize the human intestinal epithelium. The *pcnB*-encoded enzyme adds poly-A tails to the 3'-end of several transcripts [35]. Although a full picture of polyadenylation impact in bacteria is still missing, it has been linked to RNA decay in several instances, including the degradation of antisense RNAs controlling replication of R1 and ColE1 plasmids [36–38]. Here we show that the *pcnB* effect on *Shigella* growth and epithelial colonization is driven by a global positive effect on virulence gene expression. Accordingly, the *pcnB* gene sustains pINV replication to a suitable level, favouring CopA decay, particularly of an RNase E-processed form capable of exerting pINV replication control (SL-E). Ultimately, this mechanism upholds a level of pINV that enables *Shigella* invasion of and expansion within the human intestinal epithelium.

## Results

### The *pcnB* gene impairs *Shigella* growth at 37°C, but promotes intestinal epithelial cell colonization

Virulence gene expression implies energetic and metabolic expenditures, including use of extra resources and the translation of chaperones and other regulators [1]. Therefore, loss of virulence gene expression has a beneficial effect on bacterial growth (hereafter also denoted "fitness"), as exemplified by the enhanced replication speed of *Shigella* clones showing impaired Congo Red binding (virulence off) [39]. However, mapping the global cost of virulence gene expression is challenging and it has only been achieved in a few cases [40,41].

We recently mapped the genome-wide cost of virulence gene expression in *Shigella* [33]. Among chromosomally located genes, we found that the Δ*pcnB* mutant grows significantly better at 37°C (virulence on) than at 30°C (virulence off), similarly to the Δ*virF* and Δ*virB* virulence mutants (Fig 1A). In line with this, a clean reconstructed Δ*pcnB* mutant grown at 37°C showed visibly larger colonies and a reduced Congo Red binding ability compared to the wt strain, indicating faster growth and reduced virulence (S1A Fig). To validate these findings we employed our recently developed barcoded competition assay, encompassing defined consortia of genetically barcoded wt and mutant *Shigella* strains [33,42,43]. Each strain harbours a unique nucleotide barcode that can be quantified by qPCR in genomic DNA extracted from the "input consortium" and the "output consortium" following growth under specific condition(s), or following an infection assay. We here employed four replicates of a mixed 1:1:1:1:1:1 consortium, containing six strains: two wt (tagA, tagB), two Δ*mxiD* (tagC, tagD) and two Δ*pcnB* (tagE, tagF) strains. The Δ*mxiD* mutants, lacking one of the structural components of the T3SS and therefore non-invasive, represent internal controls. All strains were equally represented in the input consortium (Fig 1B). We then grew these consortia overnight (ON), either at 30° or 37°C. The Δ*pcnB* mutant showed a minor growth defect at 30°C compared to the wt and the Δ*mxiD* strains, but a notable and consistent ≥2-fold increase in the relative strain abundance at 37°C (Fig 1C).

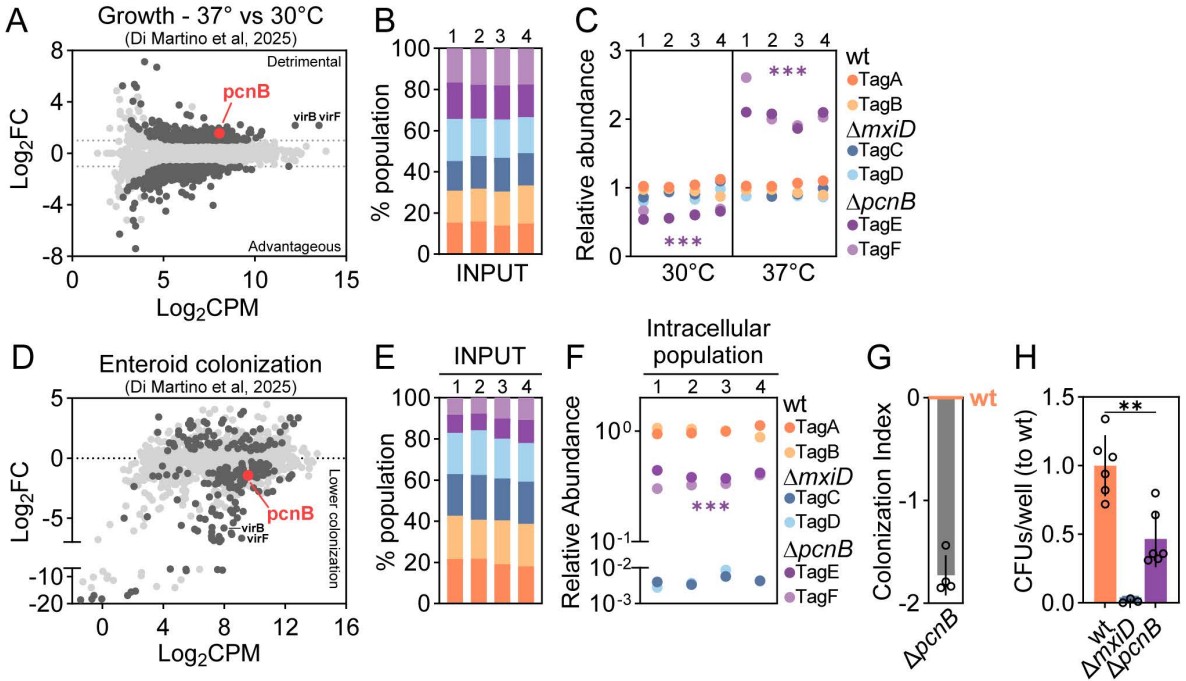

**Fig 1. *pcnB* contribution to *Shigella* fitness and colonization capacity of human intestinal epithelium.** (A) Replot from Di Martino et al, 2025 [33] - MA plot of relative mutant fitness expressed as Log$_2$FC between Growth at 37°C vs Growth at 30°C. Each dot represents a gene. Mutants with a Log$_2$FC with significant FDR (FDR ≤ 0.01) are shown in dark grey; non-significant mutants in light grey. *pcnB* mutant is highlighted in red among the detrimental genes. Log$_2$FC ≥ 1; FDR ≤ 0.01. (B-C) Barcoded competition assay in which a mixed barcoded consortium comprising two *Shigella* wt (tagA, tagB), two Δ*mxiD* (tagC, tagD) and two Δ*pcnB* (tagE, tagF) strains was grown ON at 30°C or 37°C. (B) The consortium was mixed from individual ON cultures grown at 30 °C and strain tag abundance in extracted gDNA was analysed by qPCR. The relative abundance of each strain is plotted as percentage of the total input population. (C) Quantification of relative tag abundances in the *Shigella* barcoded consortium cultures grown at 30° or 37°C, as analyzed by qPCR. Relative abundances were normalized to the input inoculum. Shown are data for four independently generated consortia. Statistical significance determined by paired t-test between the normalized output and the normalized input abundances (see Methods). ***p < 0.001. (D) Replot from Di Martino et al, 2025 [33] - MA plot showing *Shigella* gene mutant relative abundance upon infection of enteroids, as quantified by TraDIS. Shown is the Log$_2$FC in the output library on the y-axis and the Log$_2$CPM (average log$_2$ counts/million) on the x-axis. Each dot represents a gene. Mutants with a Log$_2$FC with significant FDR (FDR ≤ 0.2) are shown in dark grey; non-significant mutants in light grey. *pcnB* mutant is highlighted in red. (E-F) Enteroids were infected with a mixed barcoded consortium comprising two wt (tagA, tagB), two Δ*mxiD* (tagC, tagD) and two Δ*pcnB* mutant (tagE, tagF) strains at MOI 40 for 6h. (E) The graphs depict the composition of barcoded *Shigella* consortia used as input inocula. The relative abundance of each tag in the input consortia is plotted as percentage of the total population, as analysed by qPCR. (F) Quantification of relative tag abundance in the intracellular population, normalized against the corresponding input, as analysed by qPCR. Shown are data for four independently generated consortia for each infection. Statistical significance determined by paired t-test for each specific mutant between the normalized output and the normalized input abundances (see Methods). ***p < 0.001. (G) Bar graph showing the Colonization Index for Δ*pcnB* enteroid infections (derived from data in Fig 1E–1F and calculated as 1-(wt/mut)). wt is set as 0. (H) Intracellular *Shigella* population sizes in enteroids infected with *Shigella* wt, Δ*mxiD or* Δ*pcnB* strains for 3h at MOI 40. Results are shown as mean ± SD; CFU counts come from 6 biological replicates pooled from 2 independent experiments and were normalized to wt. Statistical significance determined by Mann-Whitney U-test; **p < 0.01.

In recent work, we also established a genome wide-map of *Shigella* genes required for the colonization of human intestinal epithelium [33]. Interestingly, in that dataset we observed a trend for the Δ*pcnB* mutant to be less capable of colonizing human epithelial enteroids (Fig 1D). Again, we employed barcoded assays to quantify this effect, mixing two wt (tagA, tagB), two Δ*mxiD* (tagC, tagD) and two Δ*pcnB* (tagE, tagF) strains and infecting enteroids at MOI 40. At 6h post infection we recovered the *Shigella* intracellular population and enriched it alongside with a dilution of the input inoculum for 14 h ON at 30°C. As expected, we again observed a slight growth defect for the Δ*pcnB* mutant after enrichment at 30°C in the input inoculum (Fig 1E). However, also after normalizing for this subtle effect, the Δ*pcnB* mutant was still ~3-fold less abundant in the intracellular population with respect to the input (Fig 1F-G), confirming a lower capacity to colonize human

enteroids. To further validate these results in the absence of secondary growth enrichment steps, we infected enteroids with the wt, the Δ*pcnB* and the Δ*mxiD* (non-invasive control) strains at MOI 40. At 3h p.i. enteroids were lysed and bacteria in the intracellular populations were quantified by plating on selective plates. This yet again revealed ~2–3 fold less CFUs for the Δ*pcnB* mutant compared to the wt strain (Fig 1H). From these data we conclude that the *pcnB* gene reduces *Shigella* fitness at 37°C, while at the same time sustains the bacterium´s ability to colonize intestinal epithelial cells. This suggests a *pcnB*-potentiating effect on *Shigella* virulence gene expression.

## The *pcnB* gene enhances T3SS expression

To shed light on the *pcnB* effect on *Shigella* virulence, we analysed expression of select virulence genes in the wt strain as well as in the Δ*pcnB* and Δ*virF* mutants by RT-qPCR (Fig 2A). As expected, all virulence genes analysed showed minimal expression levels in the Δ*virF* mutant, as well as in the wt strain grown at 30°C. In comparison, *virF*, *virB* and *icsA* expression was modestly lowered in the Δ*pcnB* mutant with respect to the wt strain grown at 37°C. However, downstream genes belonging to the T3SS-encoding mxi-spa operons, i.e. *mxiD* and *mxiE*, were ~2-fold less expressed in the Δ*pcnB* mutant than in the wt strain. This observation is compatible with the amplification cascade of *Shigella* virulence regulation, where even small changes in expression of the upstream transcriptional regulators result in significant reduction of downstream virulence gene expression [32].

To widen our understanding of the impact of *pcnB* loss on the virulence cascade, we next performed proteomic profiling. For the 2,152 identified proteins in the Δ*pcnB* mutant, 85 proteins were significantly downregulated, and 17 significantly upregulated in comparison to wt (Fig 2B; S1 Table; |log$_2$FC| ≥ 0.5; adj_p_value≤0.01). Among the downregulated proteins, 57 were proteins encoded on pINV, including the VirF and VirB transcriptional regulators, the adaptor protein for host-cell actin nucleation IcsA and nearly all T3SS structural components and secreted effectors encoded by the *ipa-mxi-spa* operons. This extends a recent observation showing that a *Shigella* Δ*ipaH2.5*Δ*pcnB* mutant secretes lower amount of the IpaB, IpaC and IpaD proteins [44]. In the Δ*virF* mutant, 74 proteins were significantly downregulated, and 10 significantly upregulated in comparison to wt (Fig 2C; S1 Table; |log$_2$FC| ≥ 0.5; adj_p_value≤0.01). Among the downregulated proteins, 48 were virulence proteins encoded on pINV, in agreement with previous reports. Overall, *pcnB* confers a similar pattern of attenuation of the virulence-relevant proteome as the master regulator VirF, although at a lower response magnitude (Fig 2B-D). Among the differences, it is further interesting to note that in the Δ*pcnB* mutant proteins with plasmid housekeeping functions were also less abundant, including the partitioning system StbA/B and the plasmid replication repressor CopB (also known as RepA2) (Fig 2D; S1 Table). This may hint to a decreased plasmid copy number, since *copB* is constitutively expressed in similar systems (e.g., R1 plasmid; [26]). These data suggest that the lower epithelial colonization capacity of the Δ*pcnB* mutant is driven by global attenuation of virulence protein expression. It appeared highly plausible that this is steered by reduced pINV replication in the absence of *pcnB*.

## The *pcnB* gene governs pINV abundance at both 30° and 37°C

In *E. coli*, lack of *pcnB* results in lower copy number of R1 and ColE1 plasmids, both characterized by sRNA-dependent replication [36,37]. To assess if *pcnB* also affects *Shigella* pINV abundance, we determined the pINV relative copies per chromosome by droplet digital PCR (ddPCR) [45]. First, we optimized the ddPCR assay for *Shigella* pINV detection, validating linearity for two primer pairs located on the pINV (in the *virF* or the *ipaA* gene, respectively) and one primer pair located close to the ter region on the chromosome (in the *manA* gene) to avoid replication biases (S2A-E Fig). Second, we extracted gDNA from wt and Δ*pcnB Shigella* cultures grown either at 30°C (virulence off) or at 37°C (virulence on) in exponential phase, employing the crude bead beating extraction method. In this setting, pINV relative copies were not affected by temperature in the wt strain, with essentially identical results when using *virF* or *ipaA* primers to detect pINV (Figs 3A; S2F). No difference in pINV relative copies was found also in the hypersecreting mutant Δ*ipaD* [46] grown at different temperatures compared to the wt strain (S2G-H Fig). pINV copy number, moreover, showed some degree

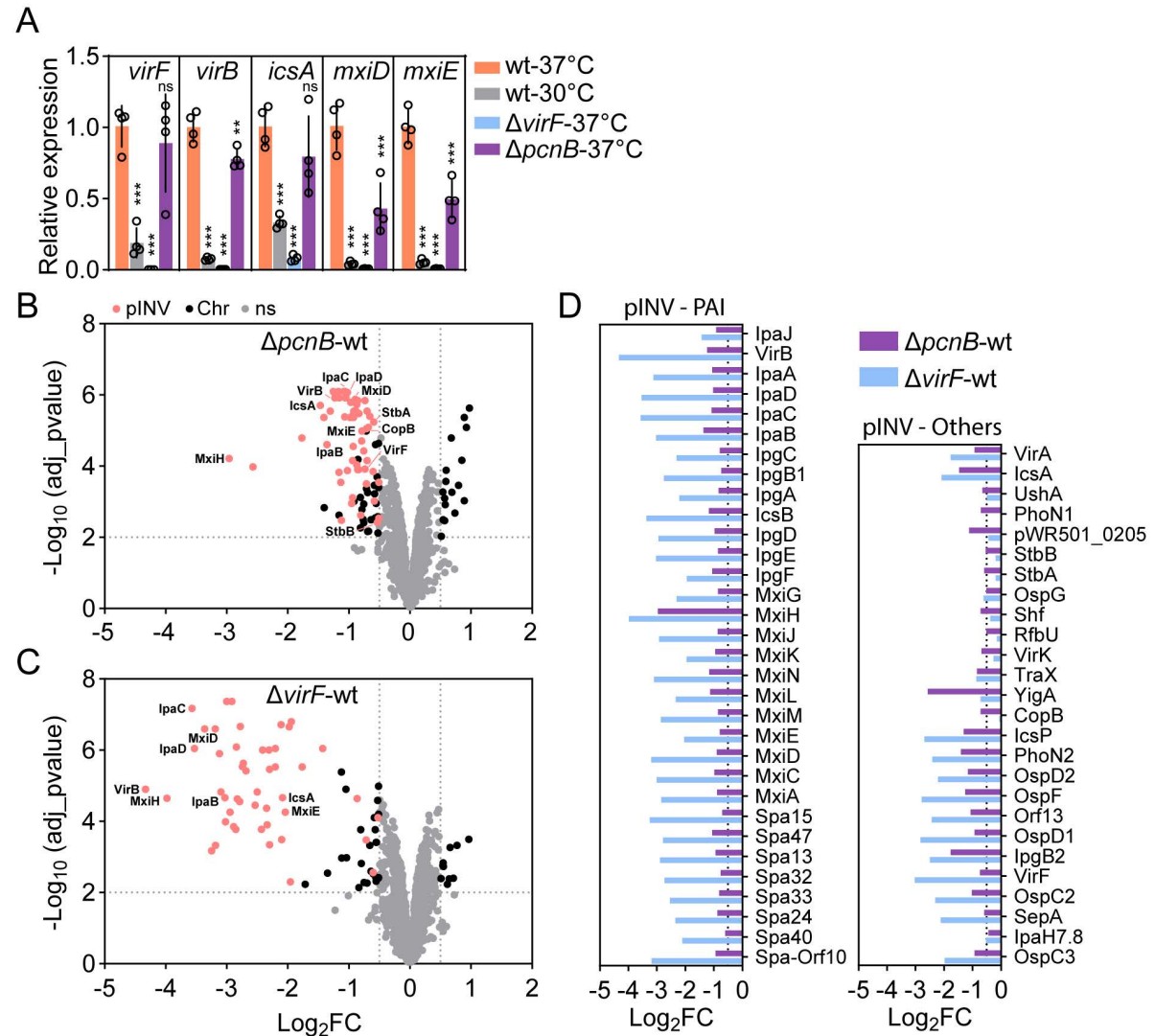

**Fig 2. Global effect of *pcnB* on pINV-encoded proteins.** (A) Expression levels of *virF, virB, icsA, mxiD* and *mxiE* analysed in *Shigella* wt strain grown at 30°C or 37°C at OD600 0.7, and in the Δ*pcnB* and Δ*virF* mutants grown at 37°C at OD600 0.7. Data were generated from four biological replicates and are presented as mean±SD. A minimum of 2 technical replicates were run in each experiment. Statistical significance determined by One-way Anova; ns - non-significant; **p<0.01, ***p<0.001. (B) Volcano plot showing differentially expressed proteins in the *Shigella* Δ*pcnB* mutant vs the wt strain. Each dot represents a protein. Differentially expressed proteins located on the pINV are shown in pink, differentially expressed proteins located on the chromosome in black, and non-significant differentially expressed proteins in grey. $Log_2FC \geq 0.5$; adj_pvalue ≤0.01. (C) Volcano plot showing differentially expressed proteins in the *Shigella* Δ*virF* mutant vs the wt strain. Each dot represents a protein. Differentially expressed proteins located on the pINV are shown in pink, differentially expressed proteins located on the chromosome in black, and non-significant differentially expressed proteins in grey. $Log_2FC \geq 0.5$; adj_pvalue ≤0.01. (D) Bar graph showing $Log_2FC$ of all differentially expressed proteins located on the pINV for the *Shigella* Δ*pcnB* vs wt and Δ*virF* vs wt comparisons.

of variation in absolute values depending on the DNA extraction method (S2I-K Fig). Most importantly, in all cases we observed a consistent ~2.5-fold reduction in pINV abundance in the Δ*pcnB* mutant compared to the wt strain, which was evident at both 30°C and 37°C (Fig 3A-B). In further agreement with this, we observed no changes in *pcnB* gene expression between the *Shigella* wt strain grown at 30°C or 37°C (Fig 3C – left panel). No changes in *pcnB* expression were also elicited by VirF expression (Fig 3C – right panel). Finally, complementation of *pcnB* in the Δ*pcnB* mutant strain restored

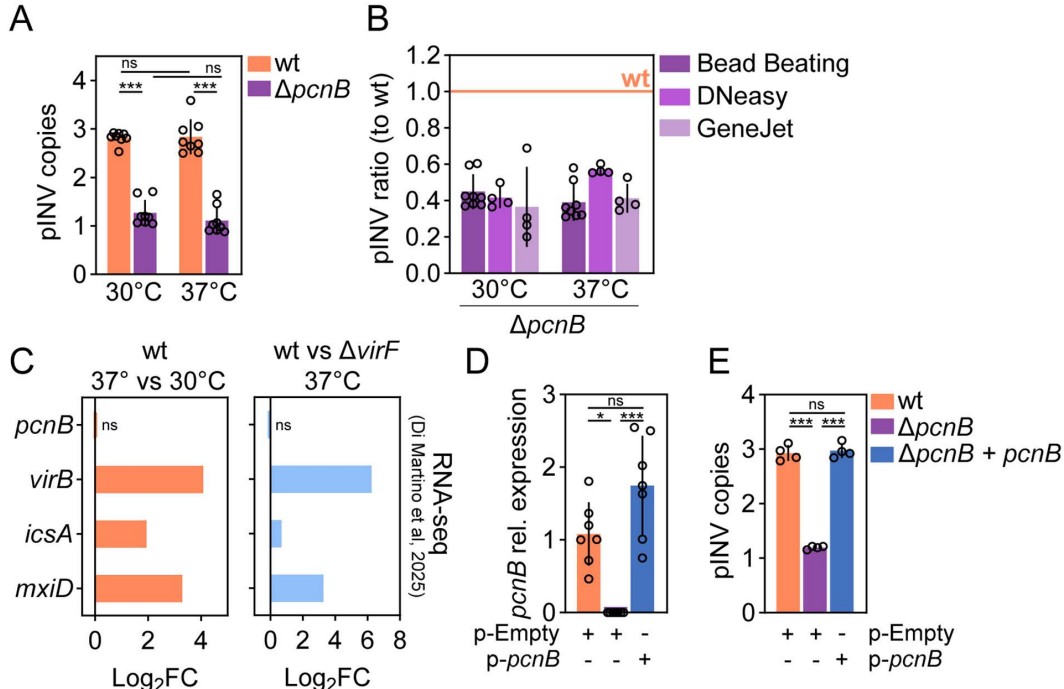

**Fig 3. *pcnB* contribution in sustaining *Shigella* pINV abundance.** (A) pINV relative copies per chromosome was determined by ddPCR using *virF* primers for pINV and *manA* primers for chromosome detection. gDNA was extracted by bead beating from *Shigella* wt or Δ*pcnB* mutant grown at 30°C or 37°C at OD600 0.7. Data comes from 8 biological replicates from two independent experiments. Statistical significance determined by Two-way Anova; ns - non-significant; ***p < 0.001. (B) pINV ratio relative to wt using gDNA from the Δ*pcnB* mutant extracted by bead beating, Qiagen DNeasy Blood & Tissue Kit or Thermo Scientific GeneJET Genomic DNA Purification Kit. pINV ratio was calculated from data from Fig 3A and S2J-K Fig. wt is set as 1. (C) Bar graphs showing RNA-seq Log$_2$FC of *pcnB* and select differentially expressed genes for the *Shigella* wt - 37° vs 30°C (right panel) and wt vs Δ*virF* – 37°C (left panel) comparisons. RNA-seq data were replotted from Di Martino et al, 2025 [33]. Significance determined by two-sided F-test with Benjamini-Hochberg correction for multiple hypothesis testing. Log$_2$FC ≥ 1; FDR ≤ 0.01. (D) *pcnB* mRNA expression levels (2$^{-\Delta\Delta Ct}$) in *Shigella* wt or Δ*pcnB* strains carrying the p-Empty or p-*pcnB* plasmid, grown at 37°C at OD600 0.7. Data come from 7 biological replicates and were normalized to the *pcnB* expression in the *Shigella* wt strain. Statistical significance determined by Kruskal-Wallis test. ns - non-significant; *p < 0.05; ***p < 0.001. (E) pINV relative copies per chromosome was determined by ddPCR using *virF* primers for pINV and *manA* primers for chromosome detection. gDNA was extracted by bead beating from *Shigella* wt or Δ*pcnB* strains carrying the p-Empty or p-*pcnB* plasmid, grown at 37°C at OD600 0.7. Data comes from 4 biological replicates. Statistical significance determined by One-way Anova; ns - non-significant; ***p < 0.001.

*pcnB* expression and also completely restored pINV copy number to wt levels (Fig 3D-E). Altogether, these data suggest that the *pcnB* gene sustains pINV replication in *Shigella* under both virulence non-inducing and inducing conditions.

### *pcnB* promotes decay of the pINV antisense RNA CopA

The *pcnB* gene encodes for the polyadenylase poly-A polymerase I (PAP-I) [34]. Mutations in *pcnB* were previously associated with the accumulation of RNase E-dependent cleavage products of antisense RNAs RNA-I and CopA, responsible for plasmid replication control of ColE1 and R1 plasmids, respectively [36,37]. These RNase E-dependent cleavage products were rapidly degraded in a wt strain (*pcnB*⁺), following polyadenylation at the 3'-end by PcnB/PAP-I activity [36]. *Shigella* CopA sRNA has 93% sequence identity with R1 CopA, suggesting a conserved RNA secondary structure (Figs 4A, S3A). Importantly, sequences important for RNase E cleavage (linker) and initial interaction with the target mRNA CopT (SL-II loop bulge) are fully conserved [47] (S3A Fig). To formally test the link between *pcnB* and pINV CopA processing and stability, we performed northern blot analysis, using a CopA probe located at the sRNA 3'end (Figs 4A, S3A). No CopA was detected in RNA extracted from an *E. coli* DH10b strain lacking the pINV. When analysing RNA extracted

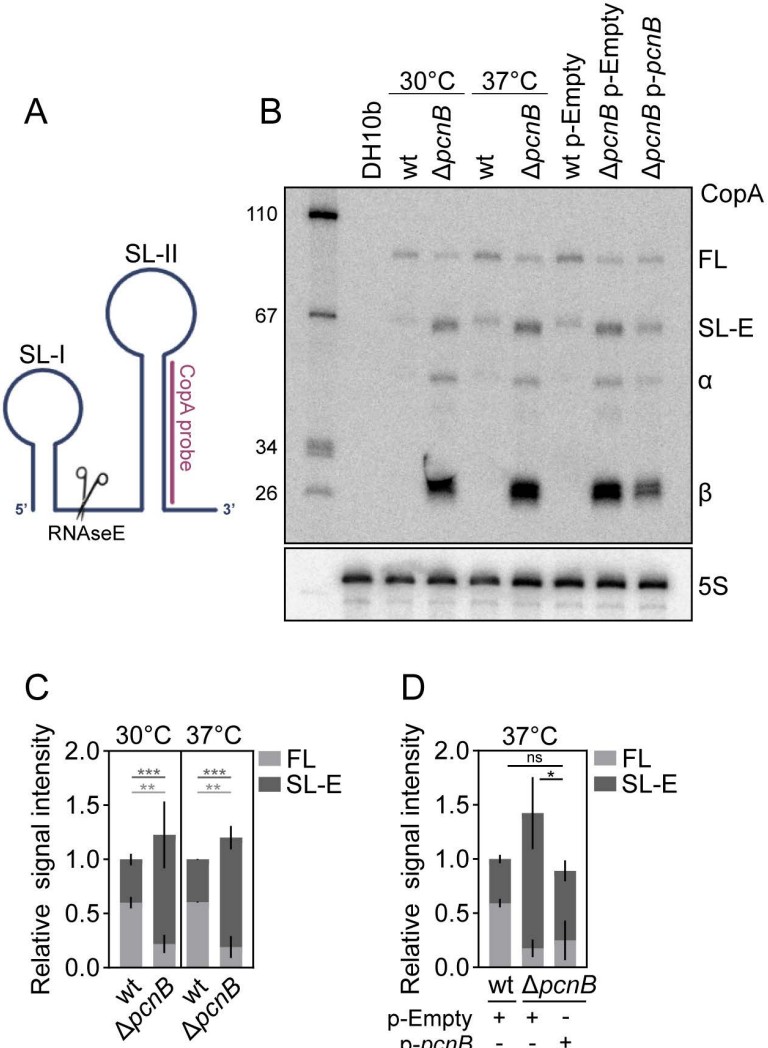

**Fig 4. Effect of *pcnB* on *Shigella* pINV CopA stability.** (A) Schematic of CopA RNA secondary structure, as resolved in [86]. Partially made with BioRender. (B) Representative northern blot of CopA species in *Shigella* wt or Δ*pcnB* mutant grown at 30°C or 37°C at OD600 0.7 and in *Shigella* wt or Δ*pcnB* strains carrying the p-Empty or p-*pcnB* plasmid, grown at 37°C at OD600 0.7. RNA from DH10b: negative control. Used as replicate 1 for quantification. (C) Relative abundance of CopA-FL+CopA-SL-E in *Shigella* wt or Δ*pcnB* mutant grown at 30°C or 37°C at OD600 0.7. CopA species abundance was determined by quantification of northern blots. Shown is the mean and SD of 3 independent experiments (See S3B Fig for replicate 2 and 3). Statistical significance determined by Two-way Anova; **p < 0.01, ***p < 0.001. Values in light grey refer to CopA-FL, in dark grey to CopA-SL-E. (D) Relative abundance of CopA-FL+CopA-SL-E in the *Shigella* wt or Δ*pcnB* strains carrying the p-Empty or p-*pcnB* plasmid was determined by quantification of northern blots. Shown is the mean and standard deviation of 3 independent experiments (See S3B Fig for replicate 2 and 3). Statistical significance for total CopA-FL+CopA-SL-E levels determined by Two-way Anova; ns - non-significant; *p < 0.05.

from the *Shigella* wt strain both at 30°C and 37°C, two major CopA species were observed: full-length CopA (CopA-FL) of ~90 nt, and CopA-SL-E, a fragment of ~65 nt previously shown to be generated by RNase E cleavage between the CopA stem loop-I and stem loop-II (Fig 4A-B). Therefore, CopA-SL-E is a processed form of CopA, which only harbours stem loop-II. Both CopA-FL and CopA-SL-E were previously found capable of inhibiting R1 plasmid replication [47,48]. However, CopA-SL-E half-life increased dramatically in the absence of *pcnB*, resulting in lower R1 copy number [36]. In our present experiments, *pcnB* deletion in *Shigella* caused a marked increase in the CopA-SL-E abundance relative to

the wt, evident at both 30 and 37°C (~2.5-fold; Fig 4B-C). This indicates that the *pcnB* encoded enzyme PAP-I favours the degradation of the *Shigella* pINV CopA-SL-E, similarly to R1 plasmid CopA-SL-E [36] (Fig 4B-C). Smaller CopA fragments (indicated as α and β) were also observed in the Δ*pcnB* mutant (Fig 4B). Likely, these forms are extremely unstable in the wt strain and therefore barely detectable. Complementation of *pcnB* was not able to restore CopA-FL relative abundance (Fig 4B-D). However, complementation fully restored the CopA-SL-E form to wt levels (Fig 4B-D), suggesting that the half-life of the CopA-SL-E transcript was restored. Notably, the total amount of CopA transcripts (FL + SL-E), both functional in replication control, matched wt levels upon complementation (Fig 4B-D), likely explaining the pINV copy number restoration (Fig 2E). Altogether, these data suggest that the *pcnB*-encoded poly-A polymerase activity targets CopA, ensuring prompt processing and degradation of the sRNA, which in turn results in suitable pINV replication levels. This warrants a beneficial trade-off between fitness and virulence, resulting in well-balanced virulence gene expression.

### *pcnB* does not affect pINV stability, but imposes virulence counter selection pressure

Our results show a clear role for the *pcnB* gene in sustaining pINV replication (Figs 3, 4). However, the reduced pINV copy number observed in the Δ*pcnB* mutant could potentially also include higher rates of plasmid loss [44]. It has been previously shown that culturing *Shigella* wt bacteria for 50 generations at 37°C results in ~60% avirulent CR- colonies. However, nearly all of them still retained pINV [49]. Instead, the CR- phenotype depended on the high plasticity of the pINV genetic architecture and was linked to the specific excision of the upstream virulence regulators *virF*, *virB* or the entire PAI loci [49]. Plasmid-free bacteria only arose in the presence of a defective MvpAT Toxin-Antitoxin (TA) system, the main TA system responsible for pINV stability at 37°C [7,49].

To assess the impact of *pcnB* on pINV stability, we introduced the *pcnB* deletion into an *ipaH9.8*-tagged strain. In this strain, the *ipaH9.8* gene, encoding for a T3SS effector located on pINV, has been disrupted by a kanamycin resistance cassette. The *Shigella ipaH9.8* mutant retains a CR+ phenotype (S4A Fig) and, as previously shown, is not significantly impaired for invasion or early intraepithelial expansion in the enteroid infection model [31]. It can therefore be used here as a reporter strain, behaving similarly to a wt *Shigella* strain during infection while providing a selectable marker on pINV. We refer to this control strain as pINV::Km. Deletion of *pcnB* in the pINV::Km strain background resulted again in a CR- phenotype (S4A Fig, compare with S1A Fig). We cultured both the control pINV::Km strain and the pINV::Km Δ*pcnB* mutant overnight at 30°C, again noting a slight growth defect for the Δ*pcnB* mutant compared to the control strain (S4B Fig, compare with Fig 1B). We then passaged the cultures daily at 37°C, a condition under which virulence gene expression is active and selection pressure for loss of virulence therefore present (Fig 5A). During early passages, the pINV::Km Δ*pcnB* mutant exhibited a growth advantage (S4B Fig; compare with Fig 1B). Interestingly, the pINV::Km Δ*pcnB* growth advantage faded over time (S4B Fig), compatible with a loss of virulence gene expression and consequently higher fitness in the control strain. We plated the inoculum as well as the passaged cultures after ~25 (p2) and ~50 (p4) generations on LB agar plates. Subsequently, 100 individual colonies per replicate were patched onto both LB and kanamycin-containing agar plates. 100% kanamycin resistant colonies were detected after 50 generations at 37°C for both the control and Δ*pcnB* stains, indicating pINV retention in either strain (Fig 5B). However, we observed the gradual emergence of bigger colonies in the pINV::Km control strain (Fig 5C and 5D; ~50% of the population after 50 generations), which likely reflected increased fitness due to loss of virulence gene expression (S4B Fig). This observation correlated with the appearance of ~50% CR- colonies after 50 generations at 37°C (Fig 5E), in full agreement with previous studies [49].

Next, we performed PCR analysis on a subset of the patched colonies after 50 generations at 37°C for the presence of pINV and the *virF* and *virB* loci. 100% of the tested colonies, derived from both control and Δ*pcnB* strains, retained pINV (Fig 5F, left panel). Notably, 100% of the big colonies from the control strain had lost either *virF*, *virB* or other loci resulting in a CR- phenotype, while small colonies retained these loci as well as the CR binding ability (Fig 5F, right panel). Finally, the vast majority of pINV::Km Δ*pcnB* colonies retained the virulence loci tested, while showing a CR- phenotype (Fig 5F, right panel).

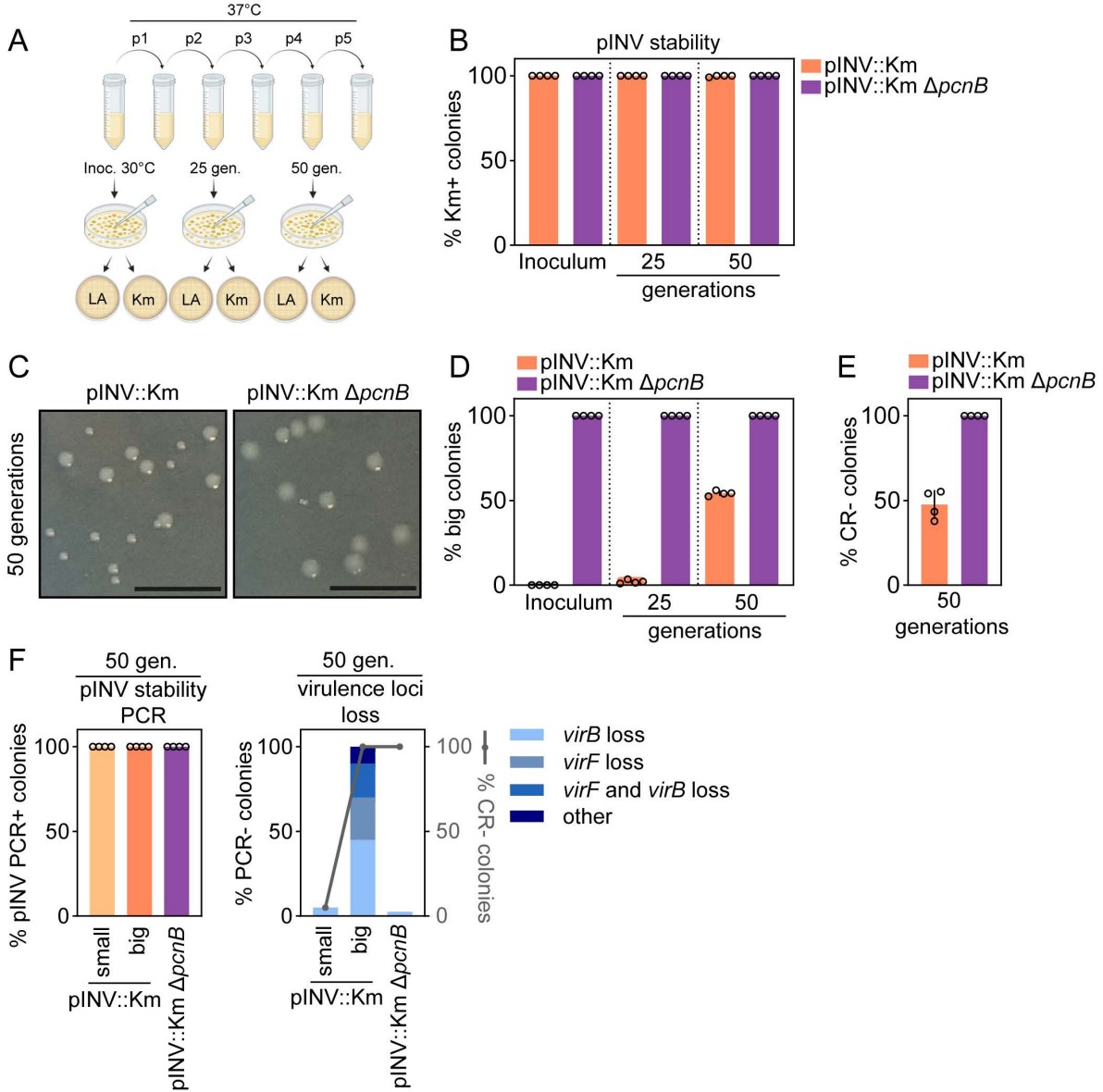

**Fig 5. The contribution of *pcnB* to pINV stability and virulence counterselection.** (A) Schematic representation of the pINV stability assay. Partially made with Biorender. (B) Stability assay of pINV plasmid in *Shigella* pINV::Km and pINV::Km Δ*pcnB* strains in the inoculum and after growth for ~25 and ~50 generations at 37˚C. The percentage of cells retaining the plasmid calculated from four biological replicates. (C) Representative images of *Shigella* pINV::Km and pINV::Km Δ*pcnB* colonies grown at 37°C for ~50 generations. Scale bar: 10mm.(D) Percentage of big colonies in *Shigella* pINV::Km and pINV::Km Δ*pcnB* strains in the inoculum and after growth for ~25 and 50 generations at 37°C. Data come from four biological replicates. (E) Percentage of CR- colonies in *Shigella* pINV::Km and pINV::Km Δ*pcnB* strains after growth for ~50 generations at 37˚C. Data come from four biological replicates. (F) Left panel. Percentage of colonies retaining pINV as scored by PCR. Right panel. Percentage of colonies lacking specified virulence loci genes. Others refers to colonies that contain *virB*, *virF* and the pINV. Twenty small colonies and 20 big colonies obtained from pINV::Km in four biological repeats and forty colonies obtained from pINV::Km Δ*pcnB* were analysed by parallel PCRs.

Altogether, these results suggest that, in the presence of active TA systems, *pcnB* does not impact pINV stability or maintenance. However, the higher rate of pINV replication sustained by *pcnB* activity increases virulence counterselection pressure under virulence inducing conditions. Upon prolonged exposure to such conditions, this ultimately results in the emergence of avirulent colonies, still harbouring pINV but lacking key virulence loci.

### The *pcnB* gene fuels *Shigella* intraepithelial expansion

As noted earlier, we observed a ~3-fold early colonization defect for the *Shigella* Δ*pcnB* mutant compared to wt (3-6h p.i.) in 3D human enteroid infections (Fig 1F–1H). Upon invasion, *Shigella* quickly escapes the endocytic vacuole and gains access to the cytosol [4]. Subsequently, *Shigella* spreads between adjacent cells via IcsA-dependent host cell actin nucleation [11], while evading innate immune signaling [31].

To follow the fate of *pcnB*-proficient and -deficient intraepithelial *Shigella* populations after intestinal epithelial cell invasion, we performed time-lapse imaging of individual invasion foci in infected enteroid-derived 2D monolayers. The approach relied on *Shigella* wt or Δ*pcnB* strains harbouring the intracellular reporter *uhpT*-GFP, as previously described [31]. The membrane-impermeable DRAQ7 dye was used as a marker for intestinal epithelial cell death and permeabilization. Strikingly, while *Shigella* wt spread efficiently within the enteroid epithelium, the Δ*pcnB* mutant completely failed to expand over time (6-20h p.i.) (Fig 6A-B; S1 Movie). To exclude a bias in fluorescent reporter intensity between the two strains, we quantified GFP mean fluorescence intensity (MFI) for each invasion focus. We observed no difference in GFP intensity between the two strains initially (4h p.i.), indicating a similar reporter intensity. Over time, GFP MFI increased for the *Shigella* wt invasion foci, but remained stable in the *Shigella* Δ*pcnB* invasion foci. This likely depends on increased number and density of intraepithelial bacteria for the *Shigella* wt invasion foci (Fig 6C). Since the live imaging setup was based on a GFP-reporter plasmid, which could potentially be affected by the lack of *pcnB*, we repeated the infections of enteroid-derived 2D monolayers with *Shigella* wt or Δ*pcnB* strains and performed immunostaining using a polyclonal anti-*Shigella* antibody. At 12 hours p.i, we again observed minimal spread of the Δ*pcnB* mutant within the intestinal epithelium, confirming and reinforcing the conclusion that the *pcnB* gene promotes *Shigella* intraepithelial expansion (Fig 6D-E). Failure at intraepithelial expansion for the Δ*pcnB* mutant could encompass multiple factors, including: i) slower/defective escape from the endocytic vacuole; ii) lower intracellular replication; iii) impaired actin-based spread, and; iv) poorer immune evasion. This aligns with the reduced expression of a broad range of virulence proteins required for the different stages of *Shigella* intracellular life in the Δ*pcnB* mutant (e.g., IpaB, IpaC and IpgD involved in vacuolar escape, IcsA involved in actin-based spread; OspC3, IpaH7.8, OspG involved in immune evasion; Fig 2B–2D; S1 Table; reviewed in [3]). Altogether, we conclude that a suitable level of pINV, sustained by *pcnB* activity, is required to raise *Shigella*'s chances of invading the intestinal epithelium and subsequently to boost intraepithelial bacterial population expansion.

## Discussion

*Shigella flexneri* navigates contrasting demands throughout infection, as virulence gene expression comes at a growth cost. Therefore, it is not surprising that expression of virulence genes, located on a large IncFII-type virulence plasmid, is strictly repressed outside the primate host. However, finding a beneficial balance between fitness and virulence is crucial for the successful colonization of the host niche. The *pcnB* gene was originally identified as a factor governing plasmid copy number of several classes of plasmids, including the IncFII R1 plasmid [36–38]. Here we show that the *pcnB* gene enables *Shigella* to fully exploit its virulence potential. In particular, we found by proteomic profiling that the *pcnB* gene enhances *Shigella* virulence gene expression at a global scale, extending recent observations [44]. This comes at a moderate cost for *Shigella* fitness. Importantly, infections of physiologically relevant human epithelial enteroid models reveal that *pcnB* boosts *Shigella* intestinal epithelial cell invasion and ultimately long-term intraepithelial expansion. Large plasmids are vital for virulence in several pathogenic bacteria and often harbour similar RepFIIA origin of replications [7]. Plasmids are usually present at an ideal copy number specific for the plasmid, the bacterial host, and in some cases for the growth condition [18,50]. This steady-state copy number usually ensures an optimal trade-off between fitness and metabolic expenditure for the expression of plasmid-encoded functions. In line with this, we collected multiple lines of evidence supporting that *pcnB* virulence-enhancing effect is achieved by sustaining *Shigella* pINV replication to this favourable level.

*pcnB* encodes the poly-A polymerase PAP-I, an enzyme that adds poly-A tails at the 3' end of a multitude of transcripts, promoting RNA degradation [34]. Similarly to the R1 plasmid, we here formally prove that the *pcnB* gene sustains *Shigella*

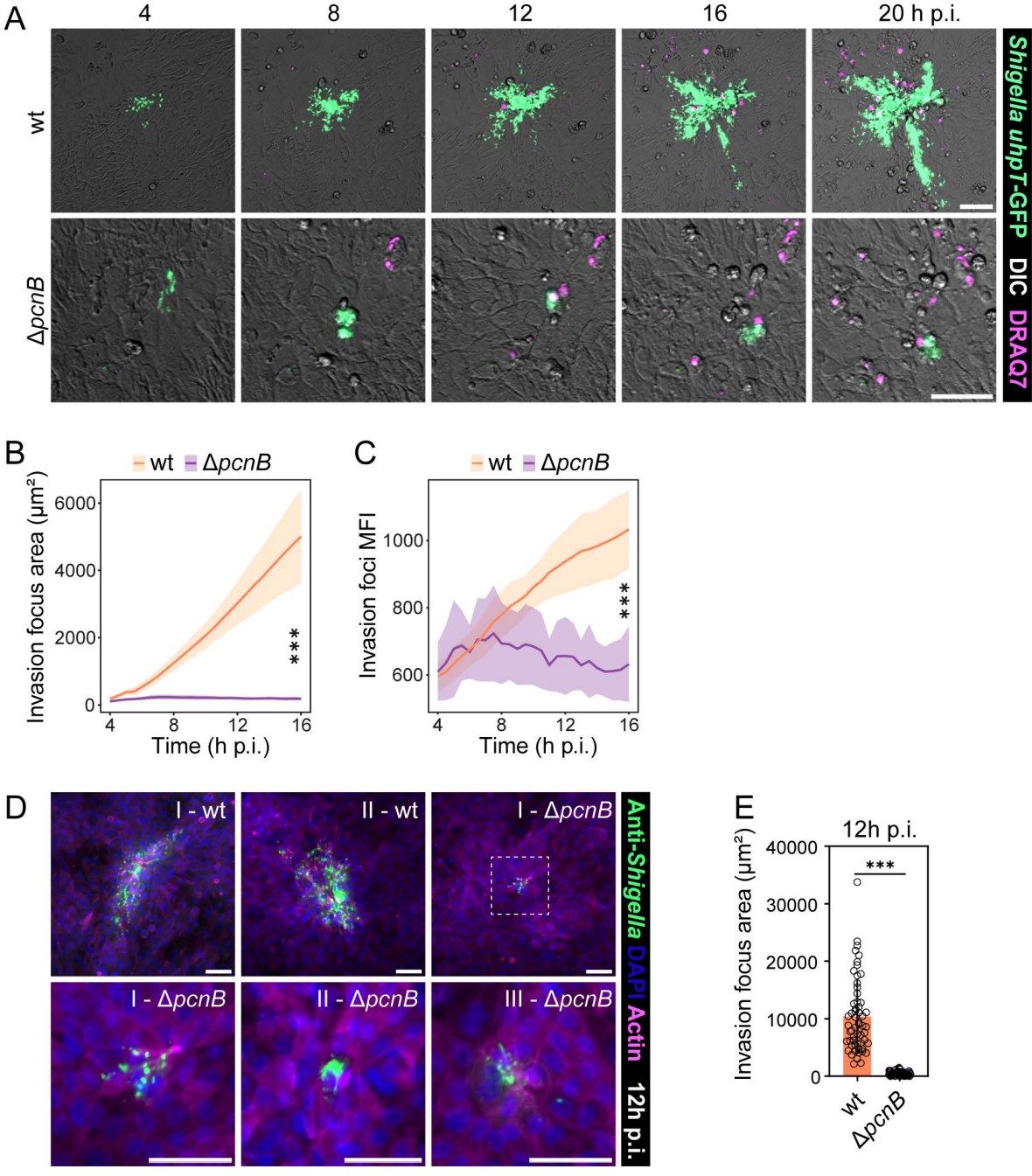

**Fig 6. The contribution of *pcnB* to *Shigella* intraepithelial expansion.** (A) Representative time-lapse series of invasion foci and DRAQ7-positive cells in enteroid-derived IEC monolayers infected with *Shigella* wt or Δ*pcnB* strains harbouring the cytosolic reporter p*uhpT*-GFP. Scale bars: 50 μm. (B) Quantification of the GFP-positive area for *Shigella* wt or Δ*pcnB* infection foci. (C) Quantification of the GFP mean fluorescence intensity over time in *Shigella* wt or Δ*pcnB* invasion foci. Plotted are mean and standard deviation for at least 6 individually infected wells per strain with at least 3 FOV (Fields of View)/well; statistical significance determined by Two-way Anova. (D) Representative fluorescent microscopy images of invasion foci in enteroid-derived IEC monolayers infected with *Shigella* wt or Δ*pcnB* strains. scale bars: 50 μm. Roman numbers represent different examples of infection foci. (E) Quantification of the GFP-positive area for *Shigella* wt or Δ*pcnB* infection foci. Plotted are mean and standard deviation of at least 50 FOVs (Fields of Views)/strain derived from 4 individually infected wells per strain; statistical significance determined by Mann-Whitney U-test; ***p < 0.001.

pINV copy number, favouring the decay of the antisense RNA CopA, which inhibits *repA* translation [36]. In particular, polyadenylation at the 3'-end of transcripts with strong secondary structures such as CopA, provides a platform for the 3′-exonucleases PNP and RNase II to initiate degradation, after cleavage by RNAse E [21]. Notably, the *pnp* gene was also reported as required for *Shigella* colonization of human intestinal epithelium [33].

Our results clearly show that the *pcnB* gene sustains pINV replication. However, the *pcnB* deletion could theoretically also result in higher rates of pINV loss. *Shigella flexneri* pINV employs two partitioning systems and three toxin-antitoxin (TA) systems to ensure plasmid maintenance in different conditions [27,51]. Partitioning systems ensure the distribution of the plasmid in daughter cells during replication. However their activity is often masked by the activity of TA systems [27]. Moreover, active TA systems rapidly kill plasmid-free bacterial cells through post-segregational killing (PSK) mechanisms [51,52]. In rare cases, pINV can also integrate into the chromosome, affecting Congo Red binding and virulence gene expression [49,53]. Notably, we showed that pINV is stably maintained in the Δ*pcnB* mutant population under virulence inducing conditions (37°C), in the presence of active partitioning and TA systems. Furthermore, the selective loss of specific virulence loci observed in the pINV::Km control strain, was abolished in the corresponding Δ*pcnB* mutant population, suggesting that the reduced pINV abundance effectively counteracts the selective pressure elicited by virulence gene expression. This, once more, highlights the fine balance between virulence, fitness and pINV dosage. While we cannot formally exclude a potential role for *pcnB* in plasmid maintenance in the absence of functional partitioning and toxin-antitoxin (TA) systems, we conclude that under physiological conditions, where these systems are active, *pcnB* does not affect pINV stability. Altogether, our results strongly suggest that *pcnB* sustains pINV replication at a level favourable for *Shigella* virulence, through the degradation of the CopA antisense RNA, rather than by explicitly favouring plasmid stability.

Pathogenic bacteria have evolved diverse mechanisms to regulate virulence gene expression in response to the host environment. In *Shigella*, sensing of the host temperature (37°C) and other host-related signals triggers the expression of the AraC-like transcriptional regulator VirF [15,16,54–56]. In a cascade-like manner, this event triggers the expression of IcsA, involved in *Shigella* intracellular spread [11,17,31], and VirB, a second transcriptional activator [30]. VirB, in turns, activates expression of the T3SS structural component and effectors, crucial for *Shigella* intestinal epithelial cell invasion and immune evasion [9]. Other bacteria have evolved different mechanisms to promote temperature-dependent virulence gene activation [57]. For example, in *Yersinia*, several RNA-Thermometers regulate the translation of the virulence master regulator LcrF [58] as well as of T3SS structural components [59,60]. Moreover, *Yersinia* increases its virulence plasmid (pYV) copy number in a temperature-dependent manner to ramp T3SS gene dosage [50,61]. pYV also harbours a Rep-FIIA origin of replication and *pcnB* deletion was recently linked to lower pYV copy number [44]. Here, for *Shigella*, we observed no obvious difference in pINV copy number in a *Shigella* wt strain grown to exponential phase at 30°C (virulence off) or 37°C (virulence on). This prompts us to propose that under the here tested conditions, *Shigella* temperature-induced virulence gene activation is achieved largely through VirF, with no or only a minor influence ascribed to pINV copy number variation.

In *E. coli*, *pcnB* levels are typically kept low across different conditions through transcriptional and translational mechanisms, with some examples for growth rate–dependent polyadenylation activity [62–66]. Our results show that, in *Shigella*, *pcnB* expression itself is not influenced by temperature, nor by VirF. Moreover, *pcnB* deletion reduced pINV copy number similarly in bacterial cultures grown at 30°C and 37°C. Therefore, it appears likely that in *Shigella pcnB* does not exert a modulatory role on pINV dosage, but rather sustains the pINV replication rate to a suitable level over a range of conditions.

Virulence is a complex trait influenced by environmental factors, the host and the pathogen. Understanding factors enhancing virulence is crucial to fight bacterial infections. Here we show that the chromosomal gene *pcnB* favours *Shigella* virulence by sustaining pINV replication. This increases *Shigella's* ability to invade human intestinal epithelial cells and boosts subsequent intraepithelial bacterial population expansion. This study provides an example of crosstalk

between genetic elements located on the chromosome and on the pINV, which results in a favourable trade-off between fitness and virulence. Ultimately, this crosstalk empowers *Shigella* to fully leverage its virulence potential.

## Materials and methods

### Ethics statement

Pseudonymized human adult stem cell-derived jejunal enteroids were established in earlier studies [67–69]. Intestinal tissue resected during bariatric surgery was used as starting material. Formal written consent was obtained from each donor. All procedures were approved under ethical permit number 2010–157 with addenda 2010-157-1 and 2020-05754, and permit nr 2023-01524-01, by the local governing body (Etikprövningsmyndigheten, Sweden).

### Bacterial strains, plasmids and general microbiological procedures

Bacterial strains are listed in S2 Table. M90T, a *Shigella flexneri* serotype 5 strain [70] was used as wt and for mutant construction. The ΔvirF and ΔmxiD *Shigella* mutants were previously generated [71,72]. For the ΔpcnB and ΔipaD *Shigella* mutants, deletions were introduced by Lambda Red recombination in the *Shigella* wt strain. wt and ΔmxiD *Shigella* barcoded strains were previously constructed and harbour TagA-B and TagC-D integrated between genes *ybhC* and *ybhB,* respectively [33]. Each genetic tag is unique and consists of 40 nucleotides [73]. For the ΔpcnB *Shigella* barcoded mutants, the *pcnB* deletion was introduced by Lambda Red recombination in the *Shigella* wt barcoded strains harbouring TagE-F [33]. The pINV::Km *Shigella* strain (ΔipaH9.8 mutant) was previously constructed [31]. For the pINV::Km ΔpcnB *Shigella* mutant, the *pcnB* deletion, bringing a CmI cassette, was introduced by Lambda Red recombination in the *Shigella* pINV::Km strain. All newly constructed mutants were verified by PCR. p*uhpT*-GFP [31,74] carries a cytosolic fluorescent reporter (S2 Table). p-Empty carries a p15A ori and AmpR [33]. An *in vitro*-synthetized DNA fragment containing *pcnB* including its native promoter was cloned in the BamHI site of the p-Empty vector, obtaining p-*pcnB*. p-*pcnB* was designed *in silico* and purchased from GenScript. Bacteria were routinely grown in Luria Bertani broth (LB) at the indicated temperatures. When necessary, antibiotics were supplemented at the following concentrations: chloramphenicol, 12.5µg/ml; kanamycin, 50µg/ml; ampicillin, 50µg/ml. Plasmid DNA extraction, DNA transformation, electrophoresis, purification of DNA fragments and sequencing were performed as described previously [75]. PCR reactions were performed using DreamTaq DNA polymerase (Thermo Fisher Scientific, #EP0702) or Phusion DNA polymerase (Thermo Fisher Scientific, #F-530L). All oligonucleotide primers and plasmids used in this study are listed in S2 Table.

### Enteroid maintenance culture

Human enteroids (pseudonym ID: 18–9jej) were previously established and thoroughly characterized elsewhere [67–69]. Human enteroids were passaged weekly by mechanical dissociation and incubation with gentle cell dissociation reagent (StemCell; #07174). Enteroid fragments were washed with DMEM-F12 (Dulbecco's modified Eagle's medium; Gibco, #11039021) containing 0.25% BSA (bovine serum albumin; Gibco; #15260-037) and re-embedded into Matrigel domes (Corning; #356230), using a 1:4 – 1:10 splitting ratio. OGM growth medium (StemCell; #06010) was exchanged every 2–4 days. Cultures were kept at 37°C in 5% $CO_2$.

### Enteroid bulk infections

Medium sized enteroids were used for bulk infections as previously described [33]. Briefly, enteroids were extracted from Matrigel domes and incubated in Cell Recovery Solution (Corning, #354253) for at least 1h on ice on a rotating table. Subsequently, enteroids were collected by sedimentation, washed with DMEM-F12/0.25% BSA, re-suspended in OGM growth medium containing 8% cold Matrigel to maintain basolateral polarity (basal-out), and aliquoted in ultra-low attachment 24-well tissue culture plates (Corning Costar; #CLS3473-24EA). Basal-out suspension cultures were

incubated at 37°C with 5% $CO_2$ for 1 day prior to infection. 1h before the infection, enteroids were washed three times with DMEM-F12/0.25% BSA, using 40µm mini cell strainers (Funakoshi, #HT-AMS-14002), re-suspended in OGM and aliquoted in ultra-low attachment 24-well tissue culture plates (Corning Costar; #CLS3473-24EA). The indicated *Shigella* strains were grown ON in LB containing appropriate antibiotics at 30°C, diluted 1:50, subcultured for 2h at 37°C without antibiotics and diluted in OGM media to obtain the desired MOI. Tagged strains were mixed in equal ratios. Bacteria were added to each well, spun down at 300g for 10min and incubated at 37°C with 5% $CO_2$. At 1h p.i., enteroids were washed 4 times with DMEM-F12/0.25% BSA using 25µm mini strainers (Funakoshi, #HT-AMS-12502) and incubated with media containing 200µg/ml gentamicin for 2h. For infections longer than 3h, media was replaced with 20µg/ml gentamicin media up to 6h p.i.. Infected enteroids were washed 6 times with DMEM-F12/0.25% BSA, recovered from the strainers and lysed in 0.1% Na-deoxycholate by homogenization with a Tissue Lyser (Qiagen). For barcoded infections, recovered intracellular bacteria were enriched in LB at 30°C for 14h. A dilution of the input inoculum was enriched similarly. For CFU counting, intracellular bacterial populations were serially diluted and plated on LB plates containing appropriate antibiotics.

### Barcoded assays – tag quantification

For barcoded competition assays and barcoded infections, the indicated tagged strains were mixed in equal ratios, following ON growth at 30°C. Genomic DNA was extracted using the the DNeasy Blood & Tissue Kit (Qiagen; #69504) from the "input consortium" and the "output consortium" (following growth under the indicated conditions, or following an infection assay). qPCR was performed to quantify each tag using Maxima SYBR green/ROX qPCR master mix (2X) (Thermo Fisher Scientific, #K0222) on a CFX384 Touch Real-Time PCR Detection System (Biorad), using 9ng of gDNA and tag-specific primers (S2 Table). The relative abundance of each strain was normalized to the abundance in the inoculum. For barcoded infection assays, a Colonization Index was calculated as 1-(avg relative abundance[WT]/ relative abundance-[MUT]) as previously described [33]. Statistical significance was determined with a paired t-test between the relative input and output abundances of each strain normalized by the respective mean WT abundance across all barcoded consortium infections or competition assays.

### qRT-qPCR

Total RNA extraction was performed with acid phenol and cDNA synthesis was performed as previously described [76]. qRT-PCR was performed using Maxima SYBR green/ROX qPCR master mix (2X) (Thermo Fisher Scientific, #K0222) on a CFX384 Touch Real-Time PCR Detection System (Bio-Rad). The levels of *virF, virB, icsA, mxiD, mxiE and pcnB* transcripts were analysed using the $2^{-\Delta\Delta CT}$ (cycle threshold [CT]) method and results are reported as relative expression with respect to the reference. The gene *nusA* was used for normalization. The oligonucleotide primers used are listed in S2 Table.

### Mass spectrometry-based proteomics

*Shigella* wt, Δ*pcnB* and Δ*virF* strains were grown at 37°C until OD600 reached ~0.7. 2ml of bacterial cultures were pelleted and washed once in PBS. Four biological replicates were collected for each strain. Bacterial cells were lysed in 50µl 2% SDS and boiling at 98°C for 10min. 5µg of each sample were denatured in a final concentration of 2% SDS and 20mM TCEP. Samples were digested as previously described [77], with a modified sp3 protocol [78]. Briefly, samples were added to a bead suspension (10µg of beads; Sera-Mag Speed Beads, 4515-2105-050250, 6515-2105-050250, in 10µl 15% formic acid and 30µl ethanol) and incubated shaking for 15min at room temperature. Beads were then washed four times with 70% ethanol. Proteins were digested overnight by adding 40µl of 5mM chloroacetamide, 1.25mM TCEP, and 200ng trypsin in 100mM HEPES pH 8.5. Peptides were eluted from the beads and dried under vacuum. Peptides were then labelled with TMTpro (Thermo Fisher Scientific), pooled and desalted with solid-phase extraction using a Waters OASIS HLB µElution Plate (30µm). Samples were fractionated onto 48 fractions on a reversed-phase C18 system running

under high pH conditions, with every sixth fraction being pooled together. Samples were analysed by LC-MS/MS, using a data-dependent acquisition strategy on a Thermo Fisher Scientific Vanquish Neo LC coupled with a Thermo Fisher Scientific Orbitrap Exploris 480. Raw files were processed with MSFragger [79] against the NCBI *Shigella flexneri* 5a strain M90T genome (CP037923 and CP037924) using standard settings for TMT. Data was normalized using vsn [80] and statistical significance was determined using limma [81]. Peptide intensities stemming from the deleted genes due to peptide co-isolation were set to zero [82].

## gDNA extraction and pINV copy number determination by droplet digital PCR (ddPCR)

*Shigella flexneri* M90T wt, M90T Δ*pcnB* and M90T Δ*ipaD* strains were grown in LB broth at the indicated temperature to an OD 600 nm of ~0.7. Genomic DNA extraction was performed by bead beating and phenol/chloroform purification. Briefly, 1.5 ml of bacterial cultures were spun down and resuspend in 250 μl of a 1:1 mix of Fast lysis buffer for bacterial lysis (QIAGEN) and nuclease-free water. The resulting samples were added to 2 ml tubes containing 0.75 g of acid-washed glass beads (diameter: 212–300 μm; Sigma Aldrich, #G1277) together with 250 μl of Phenol:Chloroform:Isoamyl Alcohol (25:24:1) (Sigma Aldrich; #77617–100ML). After 2 min ice incubation, the samples were bead beated twice with the following settings: 6.5 m/s, 24x2, 20 seconds using a FastPrep-24 Classic homogenizer (MP Biomedicals, #116004500). Subsequently the lysates were spun down for 3 min at 8,000 rpm at 4°C. The aqueous phase was transferred to a clean tube, and 1 volume of Phenol:Chloroform:Isoamyl Alcohol was added to each tube. The samples were mixed briefly by vortexing and centrifuged 3 min at 13,000 rpm at 4°C. The aqueous phase containing gDNA was transferred to a clean tube and used in subsequent ddPCR quantification. When indicated gDNA extraction was performed using the DNeasy Blood & Tissue Kit (Qiagen; #69504) or the GeneJET Genomic DNA Purification Kit (Thermo Scientific, #K0721). gDNA preparations were quantified by Qubit with the dsDNA HS Assay Kit (Invitrogen, #Q32851). ddPCR to determine pINV copy number was performed with the QX200 Droplet Digital PCR System (Bio-Rad) in 25 μl reactions containing 2x EvaGreen Supermix (Biorad, #1864034), 5U of HindIII (Thermo Scientific; #ER0502), gDNA template (final concentration: 0.78pg/ μl unless otherwise indicated) and primers (200nM each). For optimization, 6.25pg/μl of gDNA extracted by bead beating from *Shigella* wt were diluted in two-fold serial dilutions (up to 0.097656pg/ μl). To minimize pipetting errors, a pre-mix containing all components except the primers was prepared for each sample, aliquoted and primers were added last to each well. Two primer pairs located on pINV and one primer pair located on the chromosome were used (primer sequence in S2 Table; primer location in S2A Fig). Droplet generation was performed using the QX200 automated droplet generator. Subsequent PCR amplification was performed with a CFX deep-well PCR (Bio-Rad) with the following settings: initial activation at 95°C for 5 min, 39 cycles of denaturation at 94°C for 30 sec, annealing and elongation at 56°C for 1 min and one cycle of stabilization at 4°C for 10 min and 90°C for 5 min. Following PCR amplification, droplets were detected with a QX200 Dropler Reader (Bio-Rad) and pINV copy number for each sample was calculated as following: nr of positive droplets with the pINV primers/ nr of positive droplets with the chromosome primers. Total droplet count of at least 15,000 was considered reliable.

## Northern blot analysis

*Shigella flexneri* M90T wt or Δ*pcnB* strains were grown in LB broth at 30°C or 37°C to an OD 600 nm of ~0.7. *Shigella* wt or Δ*pcnB* strains carrying the p-Empty or p-*pcnB* plasmid were grown at 37°C at OD600 0.7. *E.coli.* Dh10B strain, used as a negative control, was grown at 37°C to an OD 600 nm of ~0.7. Total RNA was extracted with acid phenol as previously described [33]. Northern blotting was performed as previously described [83], with the following modifications. Five micrograms of RNA were separated on denaturing gels (6% polyacrylamide, 7M Urea, 1x TBE) and the RNA was transferred to Amersham Hybond-N+ nylon membranes (Cytiva, #RPN203B). After crosslinking at 1200 mJ/cm$^2$, the membranes were equilibrated with Church buffer [84], rolling at 42°C for 45 min. Radioactively 5' end labelled and denatured DNA probes were added and the membranes were incubated rolling at 42°C over night. The membranes were washed twice with wash

buffer (2x SSC/ 0.1% SDS), and the radioactive signals were exposed to phosphor screens and detected using a Typhoon 5 phosphoimager (Cytiva). The sequences of the DNA probes used to detect *copA* mRNA (copA_Shig) and 5S rRNA (5S) are found in S2 Table. The DNA probes were labeled using T4 polynucleotide kinase (Thermo Fisher Scientific, #EK0031) and [γ-32P]-ATP (Revvity). Unincorporated label was removed using Amersham MicroSpin G-50 columns (Cytiva, #27533001).

## pINV stability assay

*Shigella flexneri* pINV::Km or pINV::Km Δ*pcnB* strains were grown in LB broth at 30°C ON. Following this initial ON growth, cultures were diluted daily 1:5000 in 10 ml of LB medium and grown at 37°C with shaking at 180 rpm. To assess plasmid stability, samples were taken after approximately 25 and 50 generations, and cells were plated on non-selective LB-agar plates. After overnight incubation at 37°C, one hundred clones were patched on non-selective or Kanamicin-supplemented LB-agar plates to determine the presence of pINV. Plasmid stability was calculated as the percentage of colonies resistant to Km relative to the total number of colonies tested. To monitor virulence loss, samples taken at 50 generations were also plated on CR plates and incubated at 37°C. To monitor virulence loci loss, 10% of the patched colonies after 50 generations were tested by PCR for the presence of pINV (*ipaH9.8*-Km locus) and the *virF* and *virB* loci. The oligonucleotide primers used are listed in S2 Table.

## Infection and live-cell imaging of 2D human enteroid-derived monolayers

Human enteroid-derived monolayers were generated on Matrigel-coated polymer coverslips (μ-Slide 8-well high; Ibidi; #80806) as previously described [31]. In brief, 7 days after passaging, enteroids were extracted from Matrigel domes using Gentle Cell Dissociation Reagent (StemCell), washed in DMEM/F12/1.5% BSA and dissociated into single cells using TrypLE (Gibco; #12605010) and vigorous pipetting. The single cell suspension was washed again with DMEM/F12/1.5% BSA, re-suspended in OGM supplemented with 10μM Y-27632 (Sigma-Aldrich; #Y0503-1MG) and seeded out at 750'000 cells/cm2 in wells pre-coated with a 1:40 Matrigel dilution in DPBS (Gibco; #14190250). The Y-27632 ROCK inhibitor was removed 2 days after seeding. Monolayers were used for infection 3 days after seeding. *Shigella* wt or Δ*pcnB* strains carrying the intracellular reporter p*uhpT*-GFP were grown ON in LB containing 50μg/ml Ampicillin at 30°C and subcultured at 37°C in LB containing 50μg/ml Ampicillin. Equal numbers of bacteria were added to each well and spun down at 700g for 10min. Monolayers were incubated at 37°C, 5% CO2 for 40min to allow bacterial invasion. Subsequently, monolayers were washed three times with DMEM/F12/400 μg/mL gentamicin, before replacing the medium with fresh OGM containing 50μg/ml Gentamicin (Sigma, #G1914) and 0.75 μM Draq7 (Invitrogen, #D15106). Imaging was performed on a custom-built microscope based on an Eclipse Ti2 body (Nikon), using a 40x/0.6 Plan Apo Lambda air objectives (Nikon) and a Prime 95B 25mm camera (Photometrics). The imaging chamber was maintained at 37°C, 5% CO2 in a moisturized atmosphere. Bright-field images were acquired using differential interference contrast (DIC), and fluorescence was imaged using the 475/34, or 648/20 excitation channels of light engine Spectra-X (Lumencor) and emission collected through quadruple bandpass filters (89402 & 89403; Chroma). Invasion foci in enteroid-derived monolayers were manually selected and images were acquired every 30 min. Image analysis was conducted in Fiji [85] by applying rolling ball background subtraction (radius 50 pixels) to the fluorescent channels and quantifying the area and mean fluorescence intensity (MFI) above a set threshold.

## Immunostaining and fluorescent microscopy

*Shigella* wt or Δ*pcnB* strains were grown ON in LB at 30°C and subcultured in LB at 37°C prior to infection of enteroid-derived monolayers, as described above. At 12h p.i. monolayers were fixed with 2% paraformaldehyde (Pierce, #28906) for 15min at 37°C and washed three times with PBS. Subsequently, monolayers were permeabilized with 0.5% Triton-X-100 (Sigma-Aldrich, #T8787) for 10 minutes at room temperature (RT) and blocked with 3% BSA (Sigma-Aldrich,

#A9418) at 4°C ON. Monolayers were stained with anti-*Shigella* antibody (1:10) (*Shigella* polyclonal antibody, Invitrogen, #PA17245) for 2h at 37°C, followed by incubation with the secondary antibody (1:200) (Goat-α-rabbit-IgG(H + L)-AF488, Invitrogen, #A11034) for 2h at RT. Finally, monolayers were stained with 4′,6-diamidino-2-phenylindole (DAPI; 1:1,000) (Sigma, #D9542) and F-actin (1:100) (phalloidin-Alexa Fluor 647, Molecular Probes, #A22287) for 45 minutes at room temperature, before washing with PBS and imaging. Imaging was performed on a custom-built microscope based on an Eclipse Ti2 body (Nikon), using 60×/0.7 and 40×/0.6 Plan Apo Lambda air objectives (Nikon) and a and a Prime 95B 25mm camera (Photometrics).

## Supporting information

**S1 Fig. Congo Red phenotype of the *Shigella* Δ*pcnB* mutant (Supporting data for Fig 1).** (A) Representative images of CR binding of *Shigella* wt, Δ*pcnB* and Δ*mxiD* strains grown at 37°C ON. The three strains were streaked on the same CR plate. Scale bar: 10mm.
(TIF)

**S2 Fig. Optimization of experimental conditions and *pcnB* effect on *Shigella* pINV abundance (Supporting data for Fig 3).** (A) Schematic representation of *Shigella* M90T pINV [6]. Ori is indicated in teal. PAI and *virF* position are indicated in red. Position of primer pairs used to detect pINV are indicated by black lines. Partially made with BioRender. (B) Standard curve for ddPCR detection of pINV using primer pair 1 targeting the *virF* gene (as in S2A Fig; see S2 Table). 6.25 pg/µl of gDNA extracted by bead beating from *Shigella* wt were diluted in two-fold serial dilutions (up to 0.097656 pg/µl) and used to generate standard curve for *virF* primer pair. $R^2$ value indicated in the top left corner of the panel. (C) Standard curve for ddPCR detection of pINV using primer pair 2 targeting the *ipaA* gene (as in S2A Fig; see S2 Table). 6.25 pg/µl of gDNA extracted by bead beating from *Shigella* wt were diluted in two-fold serial dilutions (up to 0.097656 pg/µl) and used to generate standard curve for *ipaA* primer pair. $R^2$ value indicated in the top left corner of the panel. (D) Schematic representation of *Shigella* M90T chromosome. Ori is indicated in teal; ter region is indicated in purple. *manA* gene position is indicated in blue. Position of primer pair used to detect the chromosome is indicated by black line. Partially made with BioRender. (E) Standard curve for ddPCR detection of *Shigella* chromosome using primer pair A targeting the *manA* gene (as in S2D Fig; see S2 Table). 6.25 pg/µl of gDNA extracted by bead beating from *Shigella* wt were diluted in two-fold serial dilutions (up to 0.097656 pg/µl) and used to generate standard curves for *manA* primer pair. $R^2$ value indicated in the top left corner of the panel. (F) pINV relative copies per chromosome was determined by ddPCR using *ipaA* primers for pINV and *manA* primers for chromosome. gDNA was extracted by bead beating from *Shigella* wt or Δ*pcnB* mutant grown at 30°C or 37°C at OD600 0.7. Data comes from 8 biological replicates from two independent experiments. Statistical significance determined by Two-way Anova; ns - non-significant; ***p < 0.001. (G) pINV relative copies per chromosome was determined by ddPCR using *virF* primers for pINV and *manA* primers for chromosome. gDNA was extracted by bead beating from *Shigella* wt or Δ*ipaD* mutant grown at 30°C or 37°C at OD600 0.7. Data comes from 8 biological replicates from two independent experiments. Statistical significance determined by Two-way Anova; ns - non-significant. Data for the wt strain at 30°C and 37°C are replotted form Fig 3A. (H) pINV relative copies per chromosome was determined by ddPCR using *virF* primers for pINV and *manA* primers for chromosome. gDNA was extracted by bead beating from *Shigella* wt, Δ*pcnB* and Δ*ipaD* mutants grown at 26°C at OD600 0.7. Data comes from 4 biological replicates. Statistical significance determined by One-way Anova; ns - non-significant; ***p < 0.001. (I) pINV relative copies per chromosome was determined by ddPCR using *virF* primers for pINV and *manA* primers for chromosome. gDNA was extracted by bead beating from *Shigella* wt or Δ*pcnB* mutant grown at 30°C or 37°C at OD600 0.7. Data comes from 8 biological replicates from two independent experiments. Statistical significance determined by Two-way Anova; ns - non-significant; ***p < 0.001. Same plot as in Fig 3A for comparison. (J) pINV relative copies per chromosome was determined by ddPCR using *virF* primers for pINV and *manA* primers for chromosome. gDNA was extracted with the Qiagen DNeasy Blood &

Tissue Kit from *Shigella* wt or Δ*pcnB* mutant grown at 30°C or 37°C at OD600 0.7. Data comes from 4 biological replicates. (K) pINV relative copies per chromosome was determined by ddPCR using *virF* primers for pINV and *manA* primers for chromosome. gDNA was extracted with the Thermo Scientific GeneJET Genomic DNA Purification Kit from *Shigella* wt or Δ*pcnB* mutant grown at 30°C or 37°C at OD600 0.7. Data comes from 4 biological replicates. Interpretation I-K: Using column based gDNA extraction kits resulted in lower pINV copy number, suggesting a possible selective bias during gDNA extraction.
(TIF)

**S3 Fig. Northern blot replicates to analyse the effect of *pcnB* on *Shigella* pINV CopA stability (Supporting data for Fig 4).** (A) Sequence alignment between CopA from R1 plasmid and CopA from *Shigella* pINV. Nucleotide changes are indicated in red. Sequences for the bulge of stem-loop I and stem-loop II are indicated in bold and refer to CopA-R1 secondary structure [86]. (B) Replicates 2 and 3 of quantitative northern blot of CopA species in *Shigella* wt or Δ*pcnB* mutant grown at 30°C or 37°C at OD600 0.7 and in *Shigella* wt or Δ*pcnB* strains carrying the p-Empty or p-*pcnB* plasmid, grown at 37°C at OD600 0.7.
(TIF)

**S4 Fig. Congo Red phenotype of the *Shigella* pINV::Km Δ*pcnB* mutant and growth over multiple passages (Supporting data for Fig 5).** (A) Representative images of CR binding of *Shigella* pINV::Km (Δ*ipaH9.8*) and pINV::Km Δ*pcnB* strains grown at 37°C ON. The two strains were streaked on the same CR plate. Scale bar: 10mm. (B) OD600 measurement of pINV::Km and pINV::Km Δ*pcnB* overnight (ON) cultures grown at 30°C (inoculum) and thereafter passaged daily at 37°C.
(TIF)

**S1 Table. Proteomic profiling of *Shigella* Δ*pcnB* and Δ*virF* mutants.** Log$_2$ fold-changes and adjusted_p-values for the comparative analysis of each coding gene. Supporting data for Fig 2.
(XLSX)

**S2 Table. Strains, plasmids and oligonucleotides used in this study.**
(XLSX)

**S1 Movie. Representative movie of 2D enteroid-derived monolayers infected with the *Shigella* wt or Δ*pcnB* strains containing the cytosolic reporter p*uhpT*-GFP.** Supporting movie for Fig 6.
(MP4)

## Acknowledgments

We are grateful to members of the Sellin laboratory for helpful discussion and to Helen Wang (Uppsala University) for providing support for the ddPCR experiments.

## Author contributions

**Conceptualization:** Maria Letizia Di Martino.

**Data curation:** Maria Letizia Di Martino.

**Formal analysis:** Thibault Frisch, Petra Geiser, Margarita Komi, Philip A. Karlsson, Laura Jenniches, André Mateus, Maria Letizia Di Martino.

**Funding acquisition:** Mikael E. Sellin, Maria Letizia Di Martino.

**Investigation:** Thibault Frisch, Petra Geiser, Margarita Komi, Anjeela Bhetwal, André Mateus, Maria Letizia Di Martino.

**Methodology:** Thibault Frisch, Petra Geiser, Margarita Komi, Philip A. Karlsson, Erik Holmqvist, André Mateus, Mikael E. Sellin, Maria Letizia Di Martino.

**Project administration:** Mikael E. Sellin, Maria Letizia Di Martino.

**Resources:** Lars Barquist, Erik Holmqvist, André Mateus, Mikael E. Sellin, Maria Letizia Di Martino.

**Supervision:** Mikael E. Sellin, Maria Letizia Di Martino.

**Validation:** Mikael E. Sellin, Maria Letizia Di Martino.

**Visualization:** Petra Geiser, Maria Letizia Di Martino.

**Writing – original draft:** Maria Letizia Di Martino.

**Writing – review & editing:** Thibault Frisch, Petra Geiser, Margarita Komi, Philip A. Karlsson, Anjeela Bhetwal, Laura Jenniches, Lars Barquist, Erik Holmqvist, André Mateus, Mikael E. Sellin, Maria Letizia Di Martino.

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
