## [Decision Letter · Decision Letter 0]

25 Jun 2025

PPATHOGENS-D-25-00628

The *pcnB* gene sustains *Shigella flexneri* virulence

PLOS Pathogens

Dear Dr. Di Martino,

Thank you for submitting your manuscript to PLOS Pathogens. After careful consideration, we feel that it has merit but does not fully meet PLOS Pathogens's publication criteria as it currently stands. Therefore, we invite you to submit a revised version of the manuscript that addresses the points raised during the review process.

Please submit your revised manuscript within 60 days Aug 24 2025 11:59PM. If you will need more time than this to complete your revisions, please reply to this message or contact the journal office at plospathogens@plos.org. Please include the following items when submitting your revised manuscript:

We look forward to receiving your revised manuscript.

Kind regards,

Brian Russo

Guest Editor

PLOS Pathogens

Thomas Guillard

Section Editor

PLOS Pathogens
 
Sumita Bhaduri-McIntosh

Editor-in-Chief

PLOS Pathogens

orcid.org/0000-0003-2946-9497

Michael Malim

Editor-in-Chief

PLOS Pathogens

orcid.org/0000-0002-7699-2064

**Journal Requirements:**

At this stage, the following Authors/Authors require contributions: Thibault Frisch, Petra Geiser, Margarita Komi, Philip A. Karlsson, Anjeela Bhetwal, Laura Jenniches, Lars Barquist, Erik Holmqvist, André Mateus, Mikael E. Sellin, and Maria Letizia Di Martino. Please ensure that the full contributions of each author are acknowledged in the "Add/Edit/Remove Authors" section of our submission form.

- TM on page: 11.

4) Thank you for including an Ethics Statement for your study. Please include:

i) A statement that formal consent was obtained (must state whether verbal/written) OR the reason consent was not obtained (e.g. anonymity). NOTE: If child participants, the statement must declare that formal consent was obtained from the parent/guardian.].

5) Please upload all main figures as separate Figure files in .tif or .eps format. For more information about how to convert and format your figure files please see our guidelines: 

6) We notice that your supplementary Figures are included in the manuscript file. Please remove them and upload them with the file type 'Supporting Information'. Please ensure that each Supporting Information file has a legend listed in the manuscript after the references list.

7) We note that Figures 4 and S2 are created through BioRender. Please confirm that you hold a Premium account and provide a pdf copy of the CC BY 4.0 Licence as provided by BioRender. For instructions on how to generate a CC BY 4.0 license for your figure, please see the guidelines here: https://help.biorender.com/hc/en-gb/articles/21282341238045-Publishing-in-open-access-resources. 

If you are using the free assets from BioRender, we are unable to publish these images as they are licenced under a stricter licence than CC BY 4.0. In this case we ask you to remove the BioRender images and replace them with open source alternatives.

See these open source resources you may use to replace images / clip-art:

- https://bioart.niaid.nih.gov/ 

- https://bioicons.com/

- https://healthicons.org/ 

- https://scidraw.io/

- https://reactome.org/icon-lib

- https://www.phylopic.org/images 

- https://journals.plos.org/plosbiology/article?id=10.1371/journal.pbio.3002395

**Reviewers' Comments:**

Reviewer's Responses to Questions

**Part I - Summary**

Reviewer #1: In this manuscript, Frisch and colleagues characterize the involvement of the pcnB gene in the virulence of Shigella flexneri, the etiologic agent responsible for bacillary dysentery. They show that pcnB facilitates the expression of virulence determinants, specifically those belonging to the Type III Secretion System (T3SS), which are encoded on the virulence plasmid pINV. The authors also demonstrate that pcnB helps maintain optimal pINV copy number through the degradation of the antisense RNA CopA, the principal regulator of plasmid replication. pcnB-mediated equilibrium between bacterial proliferation and virulence expression allows Shigella flexneri to invasively colonize and disseminate in human intestinal epithelial cells efficiently, as shown in this work using enteroid models.

Overall, although the basis of pcnB function and the control of plasmid replication by antisense RNA has been known for years in other model systems, this study provides a comprehensive physiological demonstration of the role of pcnB in pINV-mediated Shigella virulence. The combination of molecular analyses (plasmid copy number and CopA RNA stability), global proteomic data, and functional validation using an advanced human infection model (enterocytes) distinguishes this work from existing data and gives it good impact in the field of bacterial pathogenesis. Previously published or preliminary data cited serve as a starting or supporting point, reinforcing the relevance of this investigation.

Throughout the manuscript, data are well presented, in general, it is well written, and figures are informative and clear. The approaches used in this study are sound. The quality of the data is good, and the use of statistics is appropriate. The methodologies used are very well described, with details to reproduce the work. References are appropriate to credit previous works.

Reviewer #2: In this report, Frish et al. show that the chromosomal gene pcnB sustains an optimal virulence plasmid copy number in Shigella flexneri. The results demonstrate that plasmid copy number is critical for optimal T3SS protein expression, with a significant impact on invasion, replication and cell-to-cell spread within epithelial cells. The work is of high quality and the conclusions are solid. This reviewer has only minor comments that involve serious editing. One additional experiment is also suggested.

Reviewer #3: •Overall, this is a well-organized, logical study.

•The main finding of this study is that PcnB influences pINV copy number via alteration of active CopA levels. All other observations are the logical consequences of this regulation. Furthermore, there are no investigations into the molecular mechanism mediating PcnB regulation of CopA activity. As a result, the impact of this study is limited.

•Significance of the findings is overstated, specifically in the claim that PcnB functions to maintain “optimal levels of pINV”.

**Part II – Major Issues: Key Experiments Required for Acceptance**

Reviewer #1: 1. The authors state that formally assessing whether pcnB affects the stability (i.e., loss) of the pINV plasmid, in addition to its replication control, is complicated by potential confounding factors such as pINV chromosomal integration and its high genetic plasticity. They conclude that pINV loss in the ∆pcnB mutant is negligible under the tested conditions, based on the restoration of plasmid copy number upon pcnB complementation. Could the authors elaborate on the rationale for considering plasmid loss negligible based solely on complementation data? Would it be feasible to design a dedicated experiment to directly quantify plasmid loss in the absence of pcnB under specific conditions (e.g., prolonged growth or absence of selective pressure for the plasmid)? Such data could more formally distinguish effects on replication from those on plasmid stability.

2. The data presented in this manuscript show that pcnB complementation restores degradation of the processed CopA-SL-E form to wild-type levels but does not restore the relative abundance of the full-length CopA-FL transcript (Figure 4). The authors suggest that CopA-SL-E is sufficient for replication control. Given that CopA-FL persists at elevated levels in the ∆pcnB mutant and is not fully restored by complementation, could this have additional effects not captured by the current assays (e.g., impacts on Tap/RepA translation beyond CopA-SL-E degradation, or other RNA interactions)? Could there be additional mechanisms regulating CopA-FL abundance or function that are independent of pcnB and PAP-I, or residual degradation activity in the mutant not enhanced by pcnB? A brief discussion on the physiological significance of the incomplete restoration of CopA-FL would be helpful.

3. The discussion mentions that polyadenylation by PAP-I creates a platform for 3’ exonucleases (PNPase and RNase II) to initiate degradation after RNase E cleavage. It is also noted that pnp is required for colonization. It would be informative to experimentally test the involvement of PNPase and/or RNase II in pcnB-mediated accelerated degradation of CopA, particularly the CopA-SL-E form (for instance, by examining CopA stability in pnp or rnb (RNase II) mutants, alone or in combination with the ∆pcnB mutant).

Reviewer #2: NA

Reviewer #3: •Is the impact of PcnB on pINV copy number mediated exclusively via regulation of CopA? ie: Are PcnB-mediated alterations observed in a strain lacking CopA?

•How does PcnB modulate CopA levels/activity?

•The conclusion is drawn that PcnB enhances Shigella virulence by “sustaining Shigella pINV replication to this optimal level”. This statement implies some sort of safe-guard against too few or too many copies of pINV. The study demonstrates that without PcnB, pINV copy number decreases, it says nothing about preventing too many copies of the plasmid.

**Part III – Minor Issues: Editorial and Data Presentation Modifications**

Reviewer #1: 4. The reviewer appreciates the clear definition of the statistical significance criteria used for the proteomic analysis (|Log2FC| ≥ 0.5; adj_p_value ≤ 0.01). Could the authors provide further details on the scientific rationale behind the choice of the Log2FC threshold at 0.5, which is quite low? Were alternative thresholds (such as Log2FC ≥ 1.0) considered or evaluated, and if so, how would the results—in terms of the number and identity of differentially expressed proteins (particularly those encoded on pINV)—and their biological interpretation have changed?

5. Related to the previous point, in the Materials and Methods section, the methodology for proteomics-based mass spectrometry is clearly described. Nonetheless, the number of independent biological replicates used to generate the proteomics data presented in Figure 2B-C and Table S1 is not explicitly stated. Could you please specify how many biological replicates were analysed for each strain (wt, ΔpcnB, ΔvirF) in this experiment? This information is important to assess the robustness and statistical support of the differential expression analysis.

6. In Figure 2B and 2C, the volcano plots clearly show the differentially expressed proteins in the proteomic analysis, and the caption confirms that each point represents a protein. However, labels in the plot use names that start with a lowercase letter, a typical convention for bacterial gene names. Since the analysis is based on protein abundance and Table S1 provides data for each "coding gene", there may be a potential inconsistency between the data represented (proteins) and the labeling convention used (genes). Could the authors provide clarification on the choice of this labeling convention or, if it is protein data, consider updating the labels to reflect the standard protein naming convention (first letter capitalized)?

Reviewer #2: The introduction section is not properly referenced. One major problem is the citation of review articles, as opposed to original articles. For instance, and this is only one example among too many, when referring to VirF as a major virulence factor on line 69, citing Ref #10 is misleading and inconsiderate. Another major problem is the exactness of citations. For instance, Ref #20 does not cover the information presented from lines 91-94 (again, just one example). The authors are therefore invited to review all the information covered in the introduction section and amend the references appropriately.

It was difficult to understand how the experiments reported in Fig. 1A and 1D have been conducted. The referenced paper (Di Martino et al, provisionally accepted) is not published and there is no related information in the M&M section. This constant reference to Di Martino et al, provisionally accepted, gets increasingly problematic across the entire paper, because the authors are referring to data that are not accessible to the readers and therefore cannot be evaluated. This needs to be addressed according to the ethical standards of the publication process.

Since the role of pcnB in Shigella virulence plasmid maintenance has been suggested already (PMID: 40424556), the authors do not need Fig 1A/D as introductory materials, and could start the results section with Fig 1B and 1D as confirmatory investigation of existing and published information. Also, please associate Ref #32 to the published PLOS Pathogens paper, not the BioRxiv version.

In Fig 1B, please indicate the growth temperature of bacteria used for generating the input consortium.

Table S1. It is difficult to understand why the log2 FC for pcnB is only -1.5 when comparing a pcnB mutant, which presumably lacks the PCNB protein, to wt? Same problem with VirF. What peptides are being detected in the mutant backgrounds?

Line 177: IcsA is not an actin nucleator. It is a bacterial adaptor that recruits the nucleation promoting factor N-WASP, which then recruits and activates the host cell actin nucleator, ARP 2/3. Please correct.

Fig. 4. The blots are apparently missing in panel B (at least in the PDF version provided to this reviewer).

Fig. 5. Although the impact of pcnB on plasmid copy number and T3SS protein expression across the manuscript is rather modest (2-3 fold range), the phenotype observed in enteroids with regard to intracellular infection overtime is rather impressive. This begs the questions as to what is happening to copy number in that system. Previous work has shown that plasmid copy number increases dramatically in vivo during Yersinia infection (PMID: 27365311). Similarly, the authors apparently have the tools to look into copy number in enteroids using qRT-PCR. That would be a fantastic addition to this work.

In general, the discussion section reads like a review of the literature with distant, direct relevance to the topic of the paper.

Reviewer #3: •Insufficient detail is given regarding the strains used in the barcoded competition assays.

•It is stated that the delta mxiD strain would express virulence genes in the absence of a functional TTSS. While some virulence genes will be expressed, expression of the entire suite of Shigella virulence genes will not be like that of WT if the TTSS is not functional. Functional secretion itself is a significant regulator of virulence gene expression in Shigella.

•It is assumed that organoids were incubated at 37C following infection, but this key information is not provided in the methods section.

•Statistical analyses are needed in Fig. 2A, Fig. 3C, Fig. 4C, Fig. 4D

PLOS authors have the option to publish the peer review history of their article (what does this mean?). If published, this will include your full peer review and any attached files.

Reviewer #1: No

Reviewer #2: No

Reviewer #3: No

**Figure resubmission:**
---

## [Decision Letter · Decision Letter 1]

14 Nov 2025

Dear Dr. Di Martino,

We are pleased to inform you that your manuscript 'The *pcnB* gene sustains *Shigella flexneri* virulence' has been provisionally accepted for publication in PLOS Pathogens.

Best regards,

Thomas Guillard, PharmD, PhD

Section Editor

PLOS Pathogens

Thomas Guillard

Section Editor

PLOS Pathogens

Sumita Bhaduri-McIntosh

Editor-in-Chief

PLOS Pathogens

orcid.org/0000-0003-2946-9497

Michael Malim

Editor-in-Chief

PLOS Pathogens

orcid.org/0000-0002-7699-2064

Reviewer Comments (if any, and for reference):

Reviewer's Responses to Questions

**Part I - Summary**

Reviewer #1: (No Response)

Reviewer #2: (No Response)

Reviewer #3: This study establishes that PcnB, a poly-A polymerase, functions to ensure sufficient copy number of the Shigella virulence plasmid (pINV). Specifically, PcnB promotes the degradation of CopA, a regulatory sRNA that represses pINV replication.

In addition to establishing this regulatory network, the presented studies nicely demonstrate that the regulatory activity of PcnB influences Shigella virulence with the use of an enteroid infections model, a strength of this investigation.

The authors were highly responsive to the first round of critiques, and the resulting resubmission is significantly stronger for that reason.

**Part II – Major Issues: Key Experiments Required for Acceptance**

Reviewer #1: (No Response)

Reviewer #2: (No Response)

Reviewer #3: None

**Part III – Minor Issues: Editorial and Data Presentation Modifications**

Reviewer #1: (No Response)

Reviewer #2: The authors have addressed my comments.

Reviewer #3: Statistical analyses are indicated in the figure legend but not shown in Fig. 3C.

PLOS authors have the option to publish the peer review history of their article (what does this mean?). If published, this will include your full peer review and any attached files.

Reviewer #1: No

Reviewer #2: No

Reviewer #3: No

---

## [Editor Report · Acceptance letter]

Dear Dr. Di Martino,

We are delighted to inform you that your manuscript, "The *pcnB* gene sustains *Shigella flexneri* virulence," has been formally accepted for publication in PLOS Pathogens.

Best regards,

Sumita Bhaduri-McIntosh

Editor-in-Chief

PLOS Pathogens

orcid.org/0000-0003-2946-9497

Michael Malim

Editor-in-Chief

PLOS Pathogens

orcid.org/0000-0002-7699-2064